# A polyamine acetyltransferase regulates the motility and biofilm formation of *Acinetobacter baumannii*

Julija Armalytė [1], Albinas Čepauskas [1,2], Gabija Šakalytė [1], Julius Martinkus [1], Jūratė Skerniškytė [1], Chloé Martens [3], Edita Sužiedėlienė[1], Abel Garcia-Pino[2] ✉ & Dukas Jurėnas [1,4] ✉

*Acinetobacter baumannii* is a nosocomial pathogen highly resistant to environmental changes and antimicrobial treatments. Regulation of cellular motility and biofilm formation is important for its virulence, although it is poorly described at the molecular level. It has been previously reported that *Acinetobacter* genus specifically produces a small positively charged metabolite, polyamine 1,3-diaminopropane, that has been associated with cell motility and virulence. Here we show that *A. baumannii* encodes novel acetyltransferase, Dpa, that acetylates 1,3-diaminopropane, directly affecting the bacterium motility. Expression of *dpa* increases in bacteria that form pellicle and adhere to eukaryotic cells as compared to planktonic bacterial cells, suggesting that cell motility is linked to the pool of non-modified 1,3-diaminopropane. Indeed, deletion of *dpa* hinders biofilm formation and increases twitching motion confirming the impact of balancing the levels of 1,3-diaminopropane on cell motility. The crystal structure of Dpa reveals topological and functional differences from other bacterial polyamine acetyltransferases, adopting a β-swapped quaternary arrangement similar to that of eukaryotic polyamine acetyltransferases with a central size exclusion channel that sieves through the cellular polyamine pool. The structure of catalytically impaired Dpa$_{Y128F}$ in complex with the reaction product shows that binding and orientation of the polyamine substrates are conserved between different polyamine-acetyltransferases.

Multidrug-resistant *A. baumannii* is a notorious hospital-acquired pathogen primarily causing ventilator-associated pneumonia and sepsis. It prevails in the environment by forming biofilms and resisting desiccation[1,2]. The name *Acinetobacter* is derived from the greek words motion ("kineto") and rod ("bakter") and means non-motile bacterium.

Nevertheless, despite the lack of flagella, *A. baumannii* possesses an uncharacterized surface-associated motility, which is recognized as one of the key molecular features that promote its environmental persistence[2]. The short polyamine 1,3-diaminopropane (1,3-DAP) is produced almost exclusively by the *Acinetobacter* genus[3] and has been

[1]Institute of Biosciences, Life Sciences Center, Vilnius University, Saulėtekio av. 7, LT-10257 Vilnius, Lithuania. [2]Cellular and Molecular Microbiology, Faculté des Sciences, Université Libre de Bruxelles (ULB), Building BC, Room 1C4 203, Boulevard du Triomphe, 1050 Brussels, Belgium. [3]Centre for Structural Biology and Bioinformatics, Université Libre de Bruxelles (ULB), Bruxelles, Belgium. Building BC, Boulevard du Triomphe, 1050 Brussels, Belgium. [4]Laboratoire de Génétique et Physiologie Bactérienne, Faculté des Sciences, Université Libre de Bruxelles (ULB), 12 Rue des Profs. Jeener et Brachet, B-6041 Gosselies, Belgium. ✉e-mail: abel.garcia.pino@ulb.be; dukas.jurenas@ulb.be

linked to surface-associated motility and virulence[4,5]. However, the molecular mechanism underlying the regulation of motility by 1,3-DAP has not been investigated. One hypothesis proposed that 1,3-DAP or its derivative could function as a signaling molecule via quorum sensing[2]. Polyamines are versatile, small positively charged (polycationic) molecules, that participate in global cellular processes such as transcription, translation, cell proliferation or stress resistance[6–8]. Most importantly, in several bacterial pathogens production of polyamines has been associated with virulence[7,9,10]. The regulation of polyamine profiles in bacteria is largely attributed to polyamine acetyltransferases, typically annotated in bacteria as SpeG (for Spermidine acetyltransferase)[11–13]. Intriguingly, *Acinetobacter* species do not produce any of the common bacterial polyamines, such as spermidine, cadaverine or putrescine[3] and do not encode the orthologues of *speG* acetyltransferase genes. As a result, an alternative acetyltransferase must exist to regulate the turnover of the *Acinetobacter*-specific polyamine 1,3-DAP and convert it to its inert form. Here we have uncovered a GNAT-family acetyltransferase conserved in *A. baumannii* that is responsible for the acetylation of 1,3-DAP in *A. baumannii* in vivo and in vitro and renamed accordingly as Dpa for diamino-propane acetyltransferase. As compared to activity of other bacterial acetyltransferases Dpa shows the highest specific activity towards 1,3-DAP. Our structural work revealed that Dpa is distinct from SpeG with a topology closer to eukaryotic polyamine acetyltransferases SSAT (Spermidine/Spermine N-acetyltransaferase) and Hpa2 (Histone acetyltransferase). Structures of catalytically impaired Dpa$_{Y128F}$ bound to its substrate polyamine trapped in pre- and post-catalytic states suggest that the mode of binding and orientation of the polyamine substrate and acetyl moiety of acetyl-CoA is shared between the different polyamine acetyltransferases.

Dpa is expressed during stationary phase and is strongly upregulated in bacteria adhering to eukaryotic epithelial cells and in bacterial pellicle formed on liquid surfaces as compared to planktonic bacteria. In agreement with previous findings that suggested the 1,3-DAP involvement in motility, we show that the chromosomal deletion of *A. baumannii* 1,3-DAP acetyltransferase gene *dpa* enhances motility and negatively affects biofilm formation on plastic. These results indicate that acetylation of 1,3-DAP is important in both, biofilm formation on abiotic surfaces and in adherence to eukaryotic cells during infection.

## Results

### Biofilm formation and cell motility are controlled by Dpa
The short polyamine 1,3-DAP has been previously associated with cell motility in *Acinetobacter*[4], however little is known about its regulation. In *Escherichia coli* and *Shigella* spp. the levels of polyamines are controlled by their acetylation thus converting them to an inert form[11,12]. We looked for acetyltransferase enzymes that could perform this function in *Acinetobacter* genus. However, a standard BLAST search of acetyltransferase homologous to the well-characterized *E. coli, Vibrio cholerae, Yersinia pestis* or *Staphylococcus aureus* polyamine acetyltransferases SpeG did not retrieve any candidates. We compared the sequences of putative GNAT proteins from *Acinetobacter* without predicted function and found distant but significant homology of one such candidate, previously annotated as CheA (renamed here as Dpa), to SpeG (16,8% identity) or eukaryotic SSAT polyamine acetyltransferase (17.2% identity)[14–16]. BLASTN analysis of available *A. baumannii* strains (515 as of December 2022) showed that the *dpa* gene was present in all sequenced strains and its nucleotide sequence was conserved with more than 95% identity. We next constructed a genomic knockout of *dpa* to assess its role in the regulation of cell motility in the context of a potential role in the control of 1,3-DAP levels[4]. A clinical *A. baumannii* isolate, belonging to international clone ICI was chosen, as it was previously shown that the isolates belonging to this international clone possess twitching motility, as opposed to the majority of isolates of international clone ICII[17]. Indeed, *A. baumannii* deleted for *dpa* migrated further under semi-liquid agar as compared

to wild-type cells (Fig. 1a), indicating that the cellular expression of *dpa* represses a bacterial movement known as twitching motility. By contrast, when 1,3-DAP was added to the medium, twitching motility despite high variability was slightly increased for WT but not for *dpa*-deleted *A. baumannii* (Fig. 1b).

Biofilm formation is intrinsically linked to cell motility as it requires undergoing the switch between exploration and colonization[18]. Indeed, deletion of *dpa* had an opposite effect on biofilm formation than cell motility−*dpa*-mutant cells formed less biofilm and the expression of the *dpa* gene from plasmid could compensate for this effect (Fig. 1c). Overall, these results indicate a direct link between the regulation of cell motility and biofilm formation and the activity of Dpa.

### *dpa* is expressed during the stationary phase and pellicle-type biofilms
Given that the presence of *dpa* was important for the regulation of motility, we wanted to know if this gene was indeed expressed in *A. baumannii* under conditions that would trigger a motility-related response from the cells. We found that the expression of *dpa* was slightly elevated in the stationary phase (by 1.6 fold; Fig. 1d) and highly elevated in pellicle-type biofilm formed on the liquid surface (by 5.3 fold; Fig. 1e), as well as in *A. baumannii* that have adhered to eukaryotic cells as compared to the expression in the planktonic cells (by 11.2 fold; Fig. 1f). These findings suggest that Dpa is active in the static cells and triggers a decrease in cellular motility.

### Dpa is an acetyltransferase specific to a substrate in *A. baumannii*
GNATs are known to target various substrates from small molecules to proteins by transferring an acetyl moiety to free amine groups[19,20]. To probe the activity of *A. baumannii* Dpa we used [14 C]acetyl-Coenzyme A as the acetyl-donor to acetylate the cellular extract of *A. baumannii* Δ*dpa* cells. Analysis of proteins and nucleic acids revealed no signal of acetylation (Fig. 2a), suggesting they are not the target of Dpa. We then separated the metabolite pool from the macromolecules in the cellular extract of *A. baumannii* by filtering out molecules with molecular weight above a 3 kDa and assayed the activity of Dpa on this sample in the presence of [14 C]acCoA. The reaction pattern revealed by thin layer chromatography, showed a band emerging from incubation of small molecule filtrate with [14 C]acCoA and Dpa enzyme (Fig. 2a). Importantly, this product was not formed when incubating *E. coli* small molecule extract with Dpa and [14 C]acCoA (Supplementary Fig. S1). This suggests that the low molecular weight target of Dpa is produced in *A. baumannii* but not in *E. coli*.

### 1,3-DAP is a primary substrate of Dpa
1,3-DAP is specific to *Acinetobacter* genus, which lack other longer-chain bacterial polyamines, such as spermidine, cadaverine or putrescine[3]. Having found that the substrate of Dpa was a small molecule specific to *A. baumannii*, involved in the regulation of motility and biofilm, we directly used a set of polyamines as a control in the acetylation assays using Dpa and [14 C]acCoA (1,3-DAP, putrescine, cadaverine, spermidine and spermine). The analysis of the acetylation pattern of control polyamines *vs.* the Dpa-treated extract from *A. baumannii* strongly indicated that the molecule 1,3-DAP is the cellular target of the enzyme (Fig. 2b). While Dpa also acetylated other polyamines, they were not detected in the molecular extract of *A. baumannii* grown in liquid LB. Apart from spermine that repressed the twitching, the addition of other polyamines did not impact the twitching motility or biofilm formation capacity of WT or *dpa*-deficient strains (Supplementary Fig. S2). However, our in vitro data supports the possibility that Dpa could acetylate other polyamines under certain conditions, such as host polyamines encountered during the infection.

At high concentration, polyamines and histones are spontaneously acetylated nonenzymatically[21,22]. Adding polyamines in higher

quantities strongly affects the pH of the reaction. However, buffering the polyamines had a strong effect on the enzyme activity. We observed a sharp increase in Dpa activity between pH 8 and 9 (Supplementary Fig. S3). Thus, to avoid nonenzymatic acetylation, we measured the kinetics of 1,3-DAP conversion to [14 C]ac-1,3-DAP at a suboptimal pH 8.5 at 22 °C. The measured $V_{max} = 50.5 \, \mu M/min$ and $K_m = 84.9 \, \mu M$ revealed that the enzyme is highly active at large excess of 1,3-DAP (Supplementary Fig. S3). In unbuffered conditions, the excess of polyamine would also increase the pH and in turn the activity of Dpa would convert the 1,3-DAP to acetyl-diaminopropane faster and thereby regulate its levels. These results showed that Dpa specifically controls the pool of 1,3-DAP *vs.* Ac-1,3-DAP in *A. baumannii*. We used N−(3-aminopropyl)-acetamide (mono-acetylated 1,3-DAP), as reaction substrate to assess whether Dpa could produce both mono- and di-acetylated products. We found that N−(3-aminopropyl)-acetamide was a very poor substrate for Dpa (Fig. 2d). Further increase in concentration of the reaction product did not inhibit the enzyme's activity (Fig. 2d). This suggested that mono-acetylated-DAP is the major product of the reaction (Fig. 2d).

## Dpa has a narrow specificity

While Dap showed some activity on other polyamines, we wanted to characterize its substrate specificity range and to assess whether other acetyltransferases could also target 1,3-DAP. In agreement with previous reports, the well-described *E. coli* polyamine acetyltransferase SpeG acetylated the long polyamines Spermine and Spermidine, however its activity was significantly lower on short-chain polyamines, and the 1,3-DAP was the worst substrate among all tested (Fig. 2c). Previously it has been reported that *A. baumannii* encodes broad specificity GNAT family acetyltransferase with suggested similarity to yeast Hpa2[23–25]. In contrast to previous reports[24,25], we did not find any significant activity of *A. baumannii* Hpa2 on any of the polyamines tested, including 1,3-DAP, nor on any aminoglycoside antibiotics tested

(Fig. 2c, Supplementary Fig. S4). Dpa expression in *E. coli* also did not confer resistance to aminoglycoside antibiotics. In agreement with these findings, Dpa did not acetylate streptomycin, kanamycin, gentamicin, tobramycin or amikacin in vitro (Supplementary Fig. S4). The *Salmonella enterica* Aac(6')-Iy enzyme was used as a control for these experiments[26,27]. Aac(6')-Iy acetylated all the aminoglycosides tested, except streptomycin and its expression in *E. coli* conferred increased MIC to all the aminoglycosides that it could acetylate (Supplementary Fig. S4). Aac(6')-Iy could also acetylate long polyamines spermine and spermidine, but not the short ones – cadaverine, putrescine or 1,3-DAP (Fig. 2c). The structure of Hpa2Ab predicted by AlphaFold2 suggested structural similarity to *S. aureus* SACOL1063 acetyltransferase[28] (Supplementary Fig. S5). We therefore confirmed that the purified Hpa2Ab enzyme was active and similarly to its homologue could acetylate threonine and some other amino acids (Trp, Tyr, Ser, Arg, His) (Supplementary Fig. S5). Once again, Dpa had no activity on amino acids, in contrast to *E. coli* SpeG which showed some activity on the same subset of amino acids as Hpa2Ab (Supplementary Fig. S5). In addition to polyamines and amino acids, SpeG could efficiently acetylate the polyamine synthesis precursor agmatine (Supplementary Fig. S5). Altogether our findings suggested that 1,3-DAP is the main target of Dpa enzyme and that this enzyme evolved specifically to acetylate 1,3-DAP, contrasting with other GNAT enzymes subfamilies (Fig. 2c).

## The structure of Dpa revealed a β-strand swapped dimer

Dpa shared low sequence identity with other bacterial and eukaryotic polyamine acetyltransferases with the most conserved region associated to acCoA binding (Supplementary Fig. S6). To better understand its structure-function interplay, we determined the structure of Dpa bound to acCoA (Fig. 3a, b). Dpa had an overall topology common to GNAT family acetyltransferases (Fig. 3a, b)[29,30]. Unlike SpeG acetyltransferases that form higher oligomers[31–33], Dpa formed a dimer

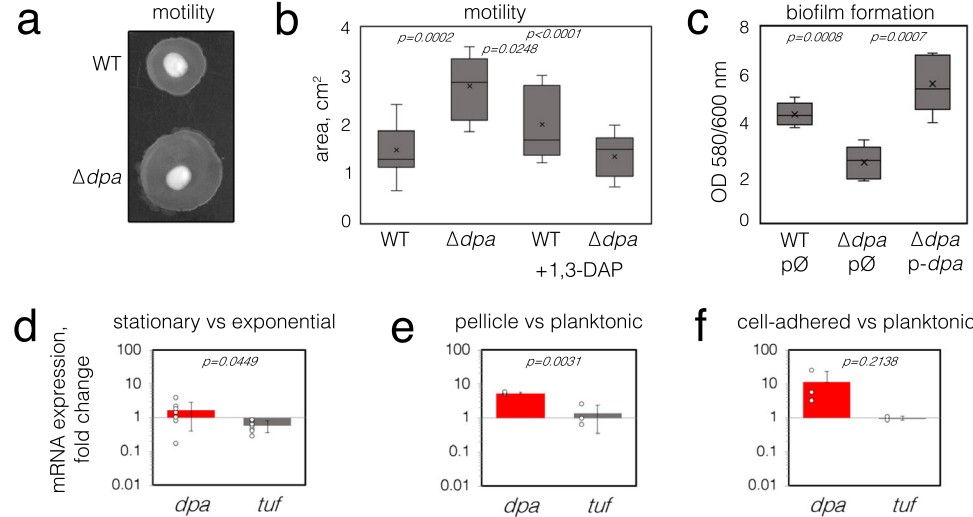

**Fig. 1 | The influence of *dpa* on cell motility and biofilm formation. a** A representative image of changes in *A. baumannii dpa* mutant twitching motility as seen on the TSB agar plate. **b** The twitching motility of *A. baumannii* was quantified by measuring the halo of growth around the inoculation site (*n* = 9) and expressed in cm². The bacteria were grown with or without 0.1 mM of 1,3-DAP added to the media. **c** The biofilm formation analysis of *A. baumannii dpa* mutant and strains with complementing plasmids. The *A. baumannii* strains with empty (*gfp* instead of *dpa*) or *dpa*-containing plasmids were inoculated to 96-well polystyrene plates, and the expression of plasmid-encoded genes was induced with 0.1 mM IPTG. Biofilm formation was measured after 18 h of growth at 37 °C by removing the planktonic cells and staining the formed biofilms with crystal violet (*n* = 5). Values were normalized to the optical density of the planktonic bacteria recovered from the well.

Top and bottom of the box plot whiskers show maximum and minimum values of the samples, top and bottom of the box − 75th percentile and 25th percentile respectively. Line through the box and x markers show the median and the mean of the sample respectively. **d**–**f** The changes in *A. baumannii dpa* expression in different conditions. The relative expression of *dpa* was analyzed by RT-qPCR, the differences in transcript amounts were evaluated by the comparative Ct method (ΔΔCt) using *rpoB* as a housekeeping gene. EF-Tu gene (*tuf*) was used as a control. Seven biological replicas were performed in (**d**) and three biological replicas were performed in (**e**, **f**). Error bars show standard deviations, two-tailed *P* values (unpaired *t*-test) are indicated above the plots. Source data are provided as a Source Data file.

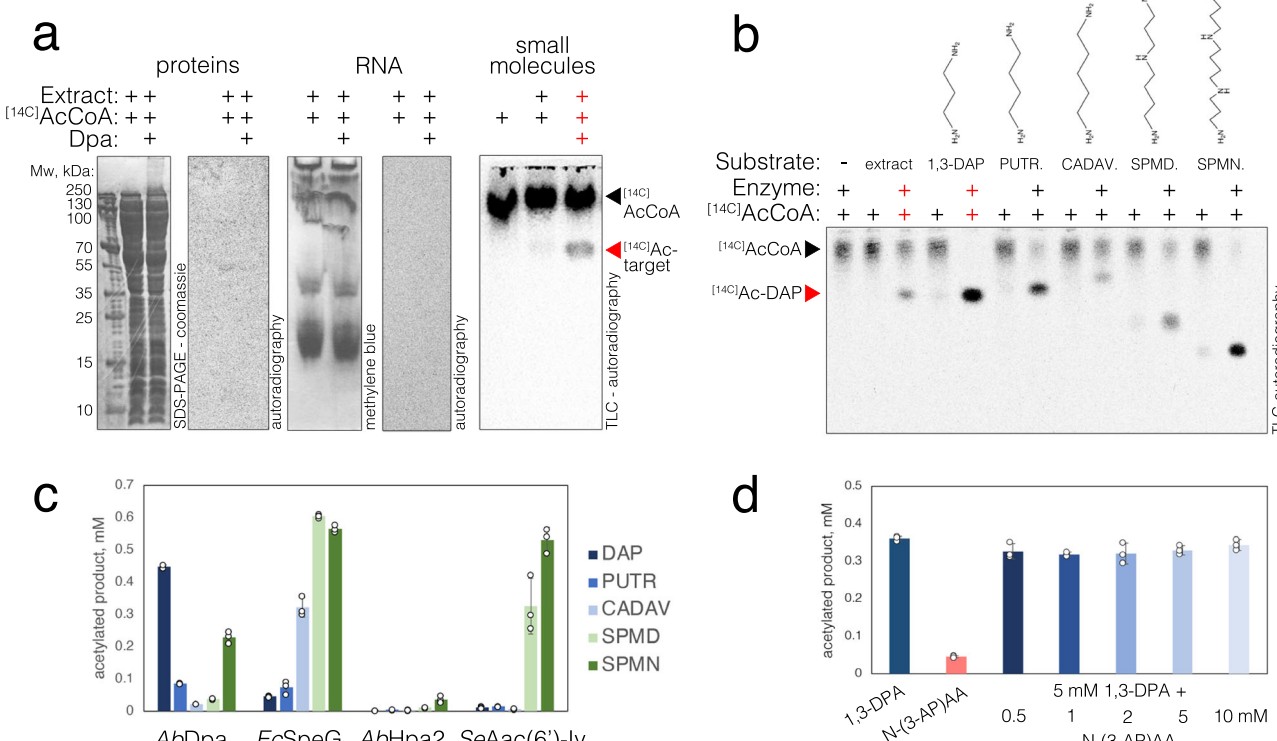

**Fig. 2 | Detection of Dpa enzyme substrate. a** Protein (left), RNA (center) and small molecule (right) extracts from exponential phase of *A. baumannii* cells cultured in liquid LB media were subjected to acetylation assays with purified Dpa enzyme and isotope labelled [14 C]acCoA. Proteins were then migrated on SDS-PAGE and stained with Coomassie; RNA was migrated on PAGE gel and stained with methylene blue. Small molecules were migrated on thin layer chromatography plate. Gels and TLC plates were dried, and isotope signals were revealed with autoradiography. Results were reproduced three times independently; representative gels and plates are shown. **b** Small molecules from *A. baumannii* cellular extract and commercial polyamine substrates: 1,3-diaminopropane (1,3-DAP), putrescine (PUTR.), cadaverine (CADAV.), spermidine (SPMD) and spermine (SPMN) were subjected to acetylation assays with purified Dpa enzyme and isotope labelled [14 C]acCoA for 30 min at 37 °C. Reactions were migrated on TLC plate and

revealed by autoradiography. Structural formulas of polyamines are indicated on top of the gel. **c** Acetylation of polyamines by bacterial GNAT acetyltransferases – *A. baumannii* Dpa, *E. coli* SpeG, *A. baumannii* Hpa2 and *S. enterica* Aac(6')-Iy. Final concentration of 5 mM of different polyamines were incubated with 0.5 mM of acCoA and 2 μM of enzymes for 30 min at 30 °C and acetylation was revealed with DTNB reagent as described in the methods section. **d** Acetylation of 1,3-DAP as compared to mono-acetylated 1,3-DAP (N-(3-aminopropyl)acetamide) indicated as N-(3-AP)AA. Reactions were performed as mentioned above, except that an increasing amount of N-(3-AP)AA was pre-mixed with the enzyme before the reaction (when indicated), otherwise 5 mM of substrates were used. Each reaction was performed three times over independent experiments, error bars show standard deviation. Source data are provided as a Source Data file.

through a C-terminal β-strand swap (Fig. 3a, b). This dimeric state was stable in solution and was not influenced by the presence of substrates acCoA and or 1,3-DAP (Supplementary Fig. S7).

This β-strand swap completed a solvent-exposed, negatively charged β-barrel-like channel that communicated the active sites of the enzyme and the acetyl-moiety of acCoA with the bulk solvent (Fig. 3a, b). The acidic nature of the channel likely facilitated the access of basic polyamines into the reaction center. Indeed, a similar acidic structure has been reported for other acetyltransferases[34–37]. The dimensions of the β-swap channel contiguous to the constricted reaction center likely favored the specificity of Dpa for the smaller 1,3-DAP polyamine. Likely, these channel features also played a role in the exit of the acetylated amines, with diminished basic properties, upon Dpa modification. Collectively these observations indicate that the β-swap interface is a distinctive functional structure of these type of GNAT polyamine acetyltransferase enzymes.

## Dpa orients polyamines similar to eukaryotic SSAT

To gain insights into the catalytic mechanism of Dpa, we used a catalytically impaired version of the enzyme carrying Y128F mutation that allowed us to determine the structure of Dpa bound to 1,3-DAP and acCoA. The crystals of Dpa$_{Y128F}$ grown in the presence of acCoA were soaked between 30 s and 1 min with 1,3-DAP before vitrification in liquid N$_2$. The compromised activity of Dpa$_{Y128F}$ (Supplementary Fig. S8)

allowed us to observe the post-catalysis state of the reaction with Ac-1,3-DAP bound to one of the two Dpa$_{Y128F}$ of the dimer (Fig. 3c, d). The other monomer of Dpa$_{Y128F}$ was bound to 1,3-DAP that was oriented slightly differently in the active site (Supplementary Fig. S9). Binding of the polyamine pre- or post-catalysis did not introduce major structural rearrangements (Supplementary Fig. S9). Interestingly, the orientation of 1,3-DAP in the active site of Dpa observed in the crystal structure (Fig. 3c, d) was similar to that observed in mouse polyamine acetyltransferase SSAT[14], suggesting that these subfamilies are functionally related.

The active center of Dpa is very acidic and involves residues Y21, Y25, Y65, Y128, D80, D81, E116 (Fig. 3d). Two densities corresponding to metal ions occupied symmetrical positions at the dimer interface and coordinate D80 and D81, damping the negative charges of the active site. These ions hold together a network of water molecules that connected and accommodated the polyamine substrate (Fig. 3e). We speculate that these ions could be Mg$^{2+}$, however we could not confirm their role in the catalysis, since prolonged incubation with EDTA did not reduce the activity of the enzyme (Supplementary Fig. S8). Nevertheless, the structure indicated a "pulling" effect of each of these ions on D80-D81 that disturbs an otherwise long β-strand introducing a bulge that is transmitted to the neighboring β-strand (Fig. 3e, f). This bulge defines the enzyme's substrate binding site and stabilizes the acetyl group of acCoA in the proper orientation (Fig. 3e, f). At the other

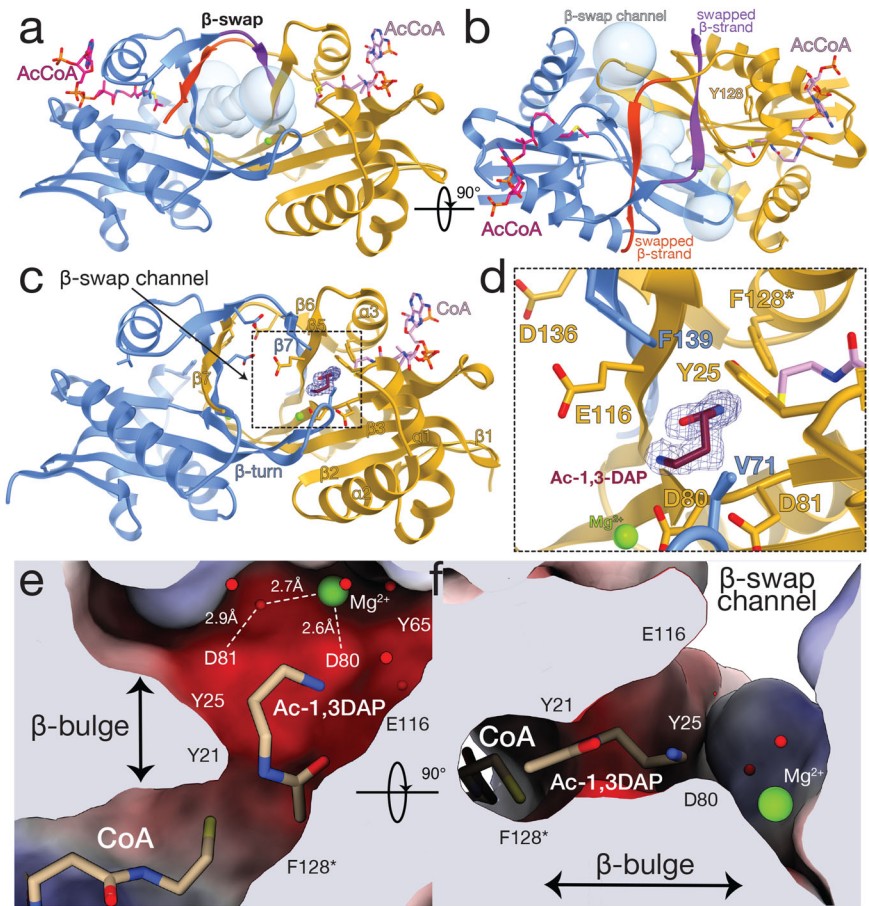

**Fig. 3 | Structure and substrate binding properties of Dpa enzyme. a** Dpa is a GNAT-fold acetyltransferase that forms a β-swapped dimer. **b** The C terminal β-strand exchange forms a channel opening to the active site residues Y128 and acetyl moiety of acCoA. **c, d** acetylation of 1,3-DAP substrate by Dpa enzyme. The contour of 2Fo-Fc map at 1.0σ of acetylated substrate is shown in blue mesh. **e, f** Mg²⁺ ion and network of water molecules at the active site of Dpa coordinate the acetylation of 1,3-DAP. The active site pocket is colored by charge distribution. Substrate-orienting residues are labeled, active site residue substitution Y128F is labeled with an asterisk.

end of the active site, 1,3-DAP is stabilized by contacts to E116 of one subunit and F139 of the other, both located at the narrow entrance of the β-swap channel, together with α1 Y25 and V71 of the β3-β4 turn. This architecture defines the pocket that accommodates the substrate and likely provides specificity to the enzyme.

In contrast to Dpa, which is a constitutive dimer (Supplementary Fig. S7), polyamines modification by SpeG seems to require conserved large oligomeric structure–polyamines bind in the interface between monomers and induce a shift of the active site loop that resounds on the overall ring-like structure and is required for catalysis[36]. In agreement with our findings using *E. coli* SpeG (Fig. 2c), it has been previously shown that *V. cholerae* SpeG could only acetylate long polyamines such as spermidine or spermine[31]. This oligomeric architecture of SpeG is directly linked to the regulation of the enzyme. It has been shown that an increase in polyamine concentration induces a structural shift that regulates the opening/closing state of the dodecameric ring required for catalysis[31,36]. While the structure of Dpa suggested that catalysis does not involve major structural rearrangements (Supplementary Figs. S7, S9), our kinetics data showed that substrate concentration, temperature as well as pH, are important for the activity of the enzyme (Supplementary Fig. S3).

## Dpa is structurally related to eukaryotic acetyltransferases Hpa2 and SSAT

Dpa is a functional dimer (Fig. 4a, b), contrasting with the large dodecameric arrangements observed in *E. coli* and *V. cholerae* SpeG

and other bacterial homologues[31,36,38]. Strikingly, Dpa has the same quaternary arrangement of a C-terminal β-strand swapped dimer of eukaryotic SSATs from mouse and human[14,34] also seen in yeast Hpa2[39] and *S. enterica* Aac(6′)-Iy[27] (Fig. 4c–e). However the yeast Hpa2 further tetramerizes despite having the overall highest structural similarity with Dpa[39] (Fig. 4e and Supplementary Fig. S6).

The SpeG dodecamers, are composed of 6 dimer units formed via a similar interface as in Dpa, but without β-strand exchange and in a slightly twisted orientation (Fig. 4b). *B. subtilis* spermine-spermidine acetyltransferase PaiA also dimerizes without β-strand swapping, which brings the two monomers closer and twists the dimer interaction surface even more as compared to Dpa (Fig. 4f). Finally, we have found that the *A. baumannii* AbHpa2 was a monomer in solution (Supplementary Fig. S7). The structural model predicted by Alpha-Fold2 suggests a compact C-terminal structure, similar to that observed in *S. aureus* SACOL1063 (Supplementary Fig. S5). Nevertheless, monomers of all these enzymes could be aligned through central β-sheet and four major α-helices (Fig. 4 and Supplementary Fig. S6).

This seven-stranded β-sheet topology of GNAT enzymes is conserved in all polyamine acetyltransferases with major topology rearrangements observed at the C-terminus. In SpeGs, as well as AbHpa2 and BsPaiA after a sharp turn, the last β-strand inserts itself between the β6 and β5 (Fig. 4b, f, g, Supplementary Fig. S6). By contrast, the last β-strand of Dpa detaches from the protein intercalating between β6 and β5 of the neighboring Dpa subunit (Fig. 4a). A comparable dimer

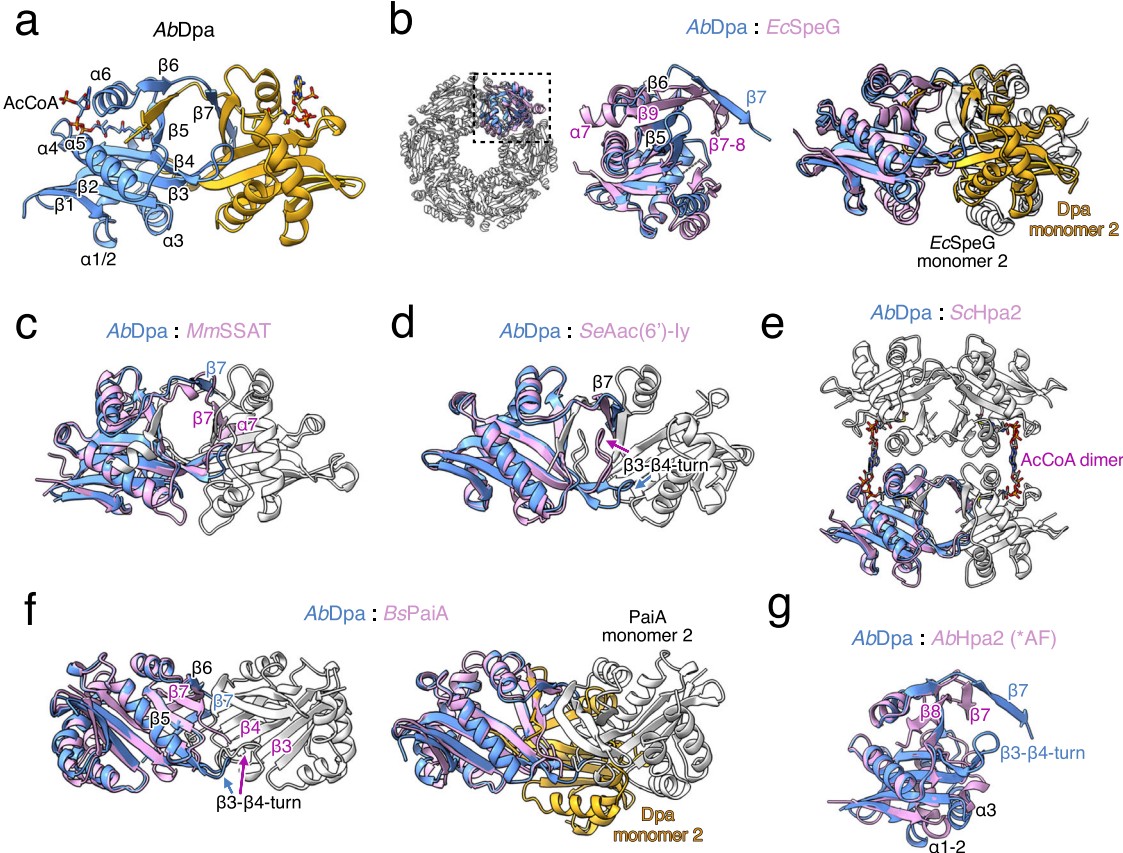

**Fig. 4 | Comparison of bacterial and eukaryotic polyamine acetyltransferases.** **a** Secondary structure elements of Dpa; α-helices and β-strands are indicated on one monomer (shown in blue). **b** Structure alignments of *A. baumannii* Dpa and *E. coli* SpeG (PDB: 4R9M) monomers (left) and dimers (right). **c** Structure alignment of Dpa with mouse SSAT (PDB: 3BJ7), (**d**) *S. enterica* Aac(6')-Iy (PDB: 1S5K), (**e**) yeast Hpa2 (PDB: 1QSO), (**f**) *B. subtilis* PaiA (PDB: 1TIQ) monomer (left) or dimer (right) and (**g**) AlphaFold2[59] model of *A. baumannii* Hpa2. Dpa monomers are colored in blue and yellow; other acetyltransferases are colored in pink for the monomer aligned and white for the rest. Sequence alignments and detailed alignments of each monomer with *r.m.s.d.* values are provided in Supplementary Fig. S6.

formation is seen in yeast Hpa2, mammalian SSATs, as well as *S. enterica* Aac(6')-Iy (Fig. 4c–e). SSATs however have an extra C-terminal α-helix that docks together with β7 against the other subunit of the dimer (Fig. 4c).

The major topological difference of polyamine acetyltransferases is dictated by the β-turn between the β-strands β3 and β4 (Supplementary Fig. S6). This variable loop defines the distance of two monomers and forms the bottom of the substrate entry channel. In dodecameric SpeGs, as well as in dimeric *Bs*PaiA and monomeric *Ab*Hpa2, this β3-β4 turn is twisted inwards (Fig. 4b, f, g, Supplementary Fig. S6). By contrast in Dpa and other dimeric GNATs the β3- β4 turn extends as far as the C terminal β-swap and together these extensions form a central substrate entrance hole (Fig. 4a, c–e). As compared to Dpa and eukaryotic polyamine acetyltransferases, the aminoglycoside acetyltransferase Aac(6')-Iy possesses very large β-turn that is flipped upwards blocking the central hole, and opening access to the active site from the other side of the turn (Fig. 4d).

### Dpa shares substrate binding properties with other polyamine acetyltransferases

The substrate-bound structures of Dpa, mouse SSAT[14] and *V. cholerae* SpeG[36] show that polyamine acetyltransferases bind their substrates through their acidic pockets. Negatively charged residues are mostly located on the β3 and β4 strands and at the inner side of α-helices α2 or α3 in the case of SpeG (Fig. 5a). Polyamines likely enter through the acidic β-swap channel of the dimeric Dpa and SSATs, while in the torus-shaped SpeGs they are suggested to travel through the central negatively charged hole of the dodecamer[36].

At the co-factor site, located opposite to the substrate-binding site, the cysteamine moiety of CoA protrudes into the catalytic center connecting with the polyamine substrate. Overall, acCoA binding is similar in all GNATs except for the coordination of the phosphoadenosine groups (Fig. 5b). In SpeG and SSAT the base is coordinated by the highly conserved P-loop sequence located between the α4 and the α5 helices and the phosphate group contacts α-helix α6 (Fig. 5b). By contrast, in Dpa, Aac(6')-Iy and PaiA the phosphoadenosine is docked entirely against α-helix α6 due to the divergent sequence of the P-loop, where the first residue in canonic [GxGx(A/G)] motif G is substituted by a large and charged reside. The large R90 sidechain in Dpa pushes the adenosine moiety against the α6 α-helix (Fig. 5b). Additionally, this orientation does not support the tetramerization seen in yeast Hpa2 which has V101 at the first position of the P-loop. The adenosine moiety of *Sc*Hpa2 is pushed to extend further out supporting the interaction with the acCoA of another monomer and thereby locking the two dimers together[39] (Fig. 5b). All these observations show that Dpa is a functionally and structurally divergent bacterial polyamine acetyltransferase, with overall stronger similarity to eukaryotic polyamine acetyltransferases SSAT or Hpa2 than to prokaryotic SpeG or PaiA.

## Discussion

The ability to form biofilms is an important adaptation for *A. baumannii* survival in hospital environments as it protects the cell from extracellular stresses and prevents penetration of antibacterial chemicals, thus increasing resistance[2,40]. Biofilm formation is also tightly connected to cell motility as a decision between sessile and motile lifestyles[18]. *A. baumannii* possesses a unique polyamine, 1,3-DAP,

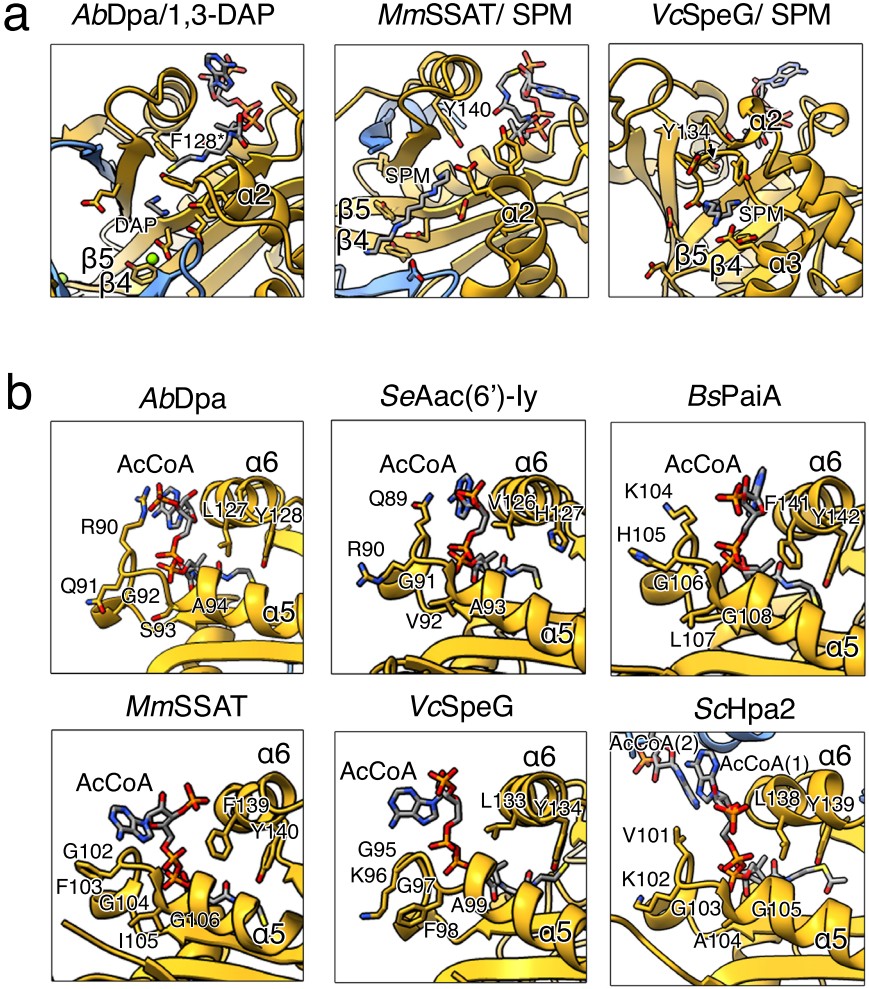

**Fig. 5 | Polyamine and acCoA binding properties of different acetyltransferases.** **a** Zoom in on the polyamine binding sites of *A. baumannii* Dpa (left), mouse SSAT (center, PDB: 3BJ8) and *V. cholerae* SpeG (right, PDB: 4R87). Acidic residues surrounding the 1,3-diaminopropane (DAP) or spermine (SPM) and catalytic tyrosine (mutated to F128 in case of Dpa, otherwise Y140 and Y134 for *Mm*SSAT and *Vc*SpeG respectively) are shown. **b** AcCoA coordination by Dpa as compared to *S. enterica* Aac(6')-ly (PDB: 1S5K), *B. subtilis* PaiA (PDB: 1TIQ), mouse SSAT (PDB: 3BJ7), *V. cholerae* SpeG (PDB: 4R87) or yeast Hpa2 (PDB: 1QSM). Conserved P-loop residues located between α-helices α4 and α5 and catalytic tyrosine or histidine residues located on the α6 helix are indicated.

synthesized from aspartate semialdehyde by the action of two enzymes - *dat* and *ddc*[41,42]. This polyamine has been associated with cell motility and was found to be important for the virulence of *A. baumannii* in the *Galleria mellonella* infection model[4]. In this study, we have found a downstream regulatory element in this pathway: the polyamine acetyltransferase Dpa that specifically acetylates 1,3-DAP in *A. baumannii* to control the effective levels of the metabolite (Fig. 6). We showed that *A. baumannii* extracts lack other polyamine acetylation signals by Dpa, suggesting that its primary substrate is 1,3-DAP. Despite the previous report that Hpa2-like enzyme could acetylate polyamines in *A. baumannii*[25], we found that it targets amino acids (Supplementary Fig. S5), and thus Dpa is so far the only *A. baumannii* polyamine acetyltransferase (Fig. 2c).

The structure of Dpa revealed a dimeric architecture with a conserved acCoA and an acidic polyamine binding site formed at the dimer interface that shares properties with other polyamine acetyltransferases. However, these enzymes diverged in topology towards the C-terminal region, which is crucial for oligomerization and substrate specificity. In Dpa, like in SSATs, a swapped dimer is formed by the β-strand exchange, while bacterial SpeG-like enzymes adopt large torus-shaped oligomeric structures. The local structure formed between the β-swapped strands and the β3-β4 turn is likely the key player defining the specificity of Dpa and regulating the size of the substrate

polyamines, which contrast with the substrate specificity of other bacterial GNAT acetyltransferases (Fig. 2c, Supplementary Figs. S4, S5). Thus, Dpa is structurally distant from the traditional bacterial polyamine acetyltransferases and closer to the well-known eukaryotic acetyltransferases SSATs and Hpa2 that acetylate polyamines[14,34,39].

Polyamines have been associated with cell motility and biofilm formation in different bacteria. In *V. cholerae*, the import of polyamines, as well as extracellular polyamines hindered biofilm formation[43]. The disruption of putrescine biosynthesis also enhanced cohesiveness and performance of *Shewanella oneidensis* biofilms and putrescine was associated with matrix disassembly[44]. Putrescine was also shown to be essential for biofilm formation in *Y. pestis*[45] but it was required for swarming motility of *Proteus mirabilis*[46]. More strikingly, in *Dickeya zeae*, putrescine was required for both cell motility and biofilm formation and the authors suggested its role as signaling molecule[47]. We have found that the expression of *dpa* increases in static *A. baumannii* cells compared to planktonic cells. Consistently, the deletion of *dpa* shows decreased biofilm formation and increased motility. This could indicate that lower 1,3-DAP concentrations are required to trigger the switch to biofilm formation. We hypothesize two different scenarios: either 1,3-DAP is required for cell motility and its acetylation shuts down the dedicated regulon, or the product of Dpa, acetyl-diaminopropane is an activator involved in biofilm formation.

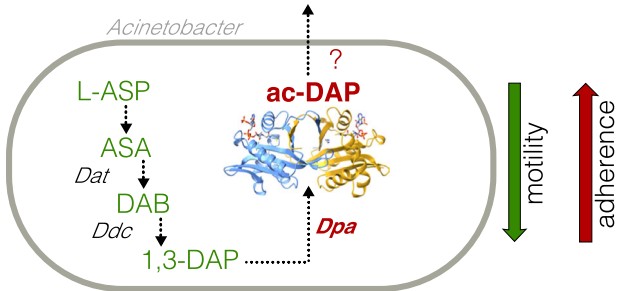

**Fig. 6 | The proposed function of the Dpa enzyme in *A. baumannii*.** 1,3-diamino propane (1,3-DAP) is synthesized from L-Aspartate (L-Asp) through L-aspartate 4-semialdehyde (ASA) and L-2,4-diaminobutanoate (DAB) intermediates, by a function of diaminobutyrate aminotransferase (Dat) and 2,4-diaminobutyrate decarboxylase (Ddc). Conserved and specific diaminopropane acetyltransferase (Dpa) acetylates and neutralizes 1,3-DAP which could be excreted. 1,3-DAP has been previously linked to surface-associated motility, while its acetylation by Dpa has the opposite effect limiting motility and increasing adherence.

While it has been shown that 1,3-DAP is required for cell motility and virulence[4], future research will be needed to assess the functionality of acetyl-diaminopropane as a potential signaling molecule. Our work shows that acetylation of 1,3-DAP plays an important role both in biofilm formation on abiotic surfaces and in adherence to eukaryotic epithelial cells. We thus anticipate that Dpa could be a suitable target for the development of novel highly specific antibacterials to combat the spread of multi-drug resistant *A. baumannii*.

## Methods

### Media and growth conditions

*E. coli* and *A. baumannii* were grown on solid and in liquid LB (Lennox L broth) medium; for twitching and pellicle formation experiments TSB (Tryptic Soy Broth, Oxoid) was used. Antibiotic concentrations were as follows—ampicillin 100 µg/ml; gentamicin 10 µg /ml. The *E. coli* DJ624Δara strain was used for cloning and protein expression. *A. baumannii* strain used in this study was isolated from clinical surgery patient and was designated to ST231 according to Oxford MLST scheme and international clone I (ICI).

### Cloning and mutagenesis

The markerless deletion of the *dpa* gene (GenBank ID: OQ718427) was performed as described previously[48], using primers presented in Supplementary Table S2. The successful deletion was confirmed by PCR.

For complementation experiments, a plasmid containing an IPTG-inducible *dpa* gene was constructed. The backbone of the previously described plasmid pUC_gm_AcORI_Ptac_gfp[49] was used, and the *gfp* gene was replaced with *dpa* (primers used to amplify *dpa* presented in Supplementary Table S2). The correct insertion of *dpa* into the plasmid was confirmed by Sanger sequencing (Baseclear).

For protein expression *dpa* and *hpa2* genes were PCR amplified from clinical *A. baumannii* isolate described previously[16], *speG* from *E. coli* K-12 MG1655, and *aac(6')-Iy* from *S. enterica* LT2 with Q5 polymerase (NEB) using primers presented in Supplementary Table S2 and cloned to pKK223.3 plasmid through EcoRI and PstI restriction sites. Amino acid sequences of all his-tagged enzymes used in this study are provided in Supplementary Table S3. Mutation in the active site of Dpa was introduced by PCR amplification of the pKK223.3-hisDpa vector with primers carrying the mutation (Supplementary Table S2). Clones were verified by Sanger sequencing (Genewiz).

### Motility, biofilm and pellicle formation and adhesion to eukaryotic cells tests

The phenotypic motility and pellicle formation tests of *A. baumannii* were performed following the protocol described earlier[17], with one

deviation that isopropanol was not used for collecting pellicles. Adhesion to mouse lung epithelial LL/2 (LLC1) cells (ATCC, CRL-1642) was also performed as described previously[17]. After infection, epithelial cells were carefully washed with DPBS three times to remove non-adherent bacteria and lysed with ddH$_2$O by intense pipetting.

### Expression quantification by RT-qPCR

*A. baumannii* cells were collected by centrifugation at 6500 *g* for 10 min at 20 °C, except for pellicles, which were removed from the 12-well plates using a sterile pipet tip and used for RNA isolation immediately. RNA was extracted using GeneJET RNA Purification Kit (Thermo Fisher Scientific), DNA removed with DNase I (Thermo Fisher Scientific), and cDNA synthesized using RevertAid First Strand cDNA Synthesis Kit (Thermo Fisher Scientific) according to the manufacturer's recommendations. qPCR was performed by CFX96 Touch Real-Time PCR Detection System (Bio-Rad) using primers presented in Supplementary Table S2 for detection of *dpa* transcripts, and for detection of *tuf* (EF-Tu) transcripts; *rpoB* was used as a house-keeping gene. For qPCR reaction (25 µl), Taq DNA polymerase (Thermo Fisher Scientific) was used according to the manufacturer's recommendations, 1.5 µM Syto9 (Invitrogen), and primer concentrations were selected according to the amplification efficiency (100% efficiency at selected concentrations). The changes in gene expression were calculated as ΔΔCt, using *rpoB* as housekeeping gene. At least three biological replicas were performed.

### Protein expression and purification

pKK223.3-his-Dpa, pKK223.3-his-Dpa$_{Y128F}$, pKK223.3-his-Hpa2$_{Ab}$, pKK223.3-his-SpeG$_{Ec}$, pKK223.2-his-Aac(6')$_{Se}$ vectors were freshly transformed into *E. coli* DJ624Δara strain. Overnight cultures were diluted 100-fold, grown until OD = 0.6 and protein synthesis was induced with 0.5 mM IPTG for 16 hours at 16 °C. Cells were then collected by centrifugation at 4000 *g* for 20 min at 4 °C, resuspended in buffer A (50 mM TRIS-HCl pH 8.5, 500 mM NaCl, 1 mM TCEP) and disrupted with a high-pressure homogenizer (Microfluidics). Protein extract was cleared by centrifugation at 20000 *g* for 30 min, filtered through a 0.45 µM filter and proteins were loaded onto HisTrapHP column (Cytiva). The column was washed with 15 CV of buffer A and proteins were eluted with a linear gradient of buffer B (50 mM TRIS-HCl pH 8.5, 500 mM NaCl, 1 mM TCEP, 1 M imidazole). Fractions containing the pure protein were pooled and concentrated using Amicon spin filter concentrator (Millipore). Protein was further purified by size exclusion gel filtration chromatography on Superdex200 10/30 column (Cytiva) pre-equilibrated with gel filtration buffer 50 mM TRIS-HCl pH 8.5, 250 mM NaCl. Protein size and purity were verified by SDS-PAGE.

### Analytic size exclusions chromatography

Analytic SEC runs were performed using Superdex75 increase 1030 column (Cytiva) pre-equilibrated with gel filtration buffer (50 mM TRIS-HCl pH 8, 250 mM NaCl) and injecting 500 µL of 20 µM of protein. When indicated, proteins were pre-incubated with 5-fold molar excess of acCoA and/or 1,3-DAP. All the proteins used for gel filtration analyses were migrated on 15% SDS-PAGE gel and stained with Coomassie.

### Protein crystallization

Purified his-Dpa protein was concentrated to 10 mg/ml using a spin filter concentrator (Amicon) and screened for crystallization with commercial screens Crystal Screen and Crystal Screen 2 (Hampton Research), JCSG-plus, PACT premier, PEGRx HT (Molecular Dimensions) in the presence of 0.1 mM acetyl-CoA. Sitting drops of 100 nl protein solution topped with 100 nl precipitant solution were equilibrated against 80 µl precipitant solution. Drops were set up using Mosquito robotic system (TTP Labtech) in Swisci 96-well 2-drop plates

and incubated at 20 °C. Crystals grown in 0.2 M Sodium acetate trihydrate, 0.1 M TRIS hydrochloride pH 8.5, 30% w/v Polyethylene glycol 4,000 were of best diffraction quality. Crystals were quick soaked in mother liquor solution supplied with a final concentration of 2 M sodium bromide and 20% PEG400 for cryo-preservation and vitrified in liquid nitrogen.

The his-Dpa Y135F protein was purified in identical conditions to the wild-type. Protein crystals were set up in 0.2 M Sodium acetate trihydrate, 0.1 M TRIS hydrochloride pH 8.5, 30% w/v Polyethylene glycol 4,000, 0.1 mM acetyl-CoA. Crystals were immersed in a cryogenic solution (mother liquor with 20% PEG400) supplemented with 2 M 1,3-DAP (Sigma) for a 30–60 s before vitrification in liquid nitrogen.

### X-ray data collection and processing

Diffraction data were collected on the PROXIMA-1 (PX1) and PROXIMA-2A (PX2A) beamlines at the SOLEIL synchrotron, Gif-sur-Yvette, Paris, France. Data were indexed, integrated with XDS[50], scaled with XSCALE[50] or AIMLESS[51], quality and twinning were assessed with phenix.xtriage[52] and POINTLESS[51]. Data from his-Dpa crystals were collected at the Br K edge for experimental phasing. Cell-content analysis was performed with MATTHEWS_COEF[53]. The heavy-atom substructure was found with ShelxD as implemented in the HKL2MAP suite[54,55].

The initial model of Dpa was built using ShelxE and subsequently used in ArpWarp[56], which generated 90% of the model. After several iterations of manual building with Coot[57] and maximum likelihood refinement as implemented in Buster/TNT[58], the model was extended to cover all the residues (R/Rfree of 18.2/20.7 %). The structure of his-Dpa$_{Y128F}$ in complex with 1,3-DAP was solved by molecular replacement using his-Dpa as the search model. After molecular replacement with as implemented in the the Phenix package[52], the model of the complex was completed by manual building with Coot[57] and maximum likelihood refinement as implemented in Buster/TNT[58] (R/Rfree of 19.6/23.2 %). All the details of the data processing and refinement are listed in Supplementary Table S1.

### Enzymatic reactions using [14 C]Ac-CoA

For protein, RNA and small molecule extract acetylation, bacterial cells were grown to the mid-exponential phase (OD = 1). Cells were then disrupted as described above in the protein purification section. Small molecules were separated from proteins by retaining the proteins in a spin filter concentrator (Amicon) with 3 kDa cut-off. The filtrate was further concentrated using a speed vacuum concentrator. RNA was extracted with standard hot-phenol and chloroform extraction procedure, then precipitated with isopropanol and dissolved in water.

All acetylation reactions were supplied with 200 µM radiolabeled [14 C]Ac-CoA (60 mCi/mmol) and 1 µM his-Dpa when indicated. Reactions were incubated at 37 °C for 30 min. For kinetics experiments reactions were incubated at 22 °C in Tris-HCl buffer at pH 9 and stopped at different times by adding 50% cold TCA.

Proteins were resolved on 4–20% gradient SDS-PAGE gel (BioRad) and nucleic acids were resolved on 10% native acrylamide (19:1) TBE (Tris-borate EDTA) gel that were stained with coomassie blue or methylene blue respectively. Acetylation reactions of small molecules and commercial polyamines (Sigma) were resolved on cellulose TLC plates in the isopropanol/HCl/water (68/18/14) mobile phase. Gels and plates were dried and exposed to multipurpose phosphor storage screen (Amersham) overnight and scanned using the Storm 860 PhosphoImager system (Molecular Dynamics). For kinetics, the ac-1,3-DAP was quantified from the band intensity measured with the ImageJ program.

### Enzymatic reactions using DTNB

Reactions were assembled in 96-well clear polystyrene flat-bottom plates. 50 mM Tris-HCl pH 8.0, 0.5 mM acCoA, 5 mM substrates, 2 µM of acetyltransferase enzymes in reaction volumes of 50 µL. Reactions were proceeded for 30 min at 30 °C, stopped by adding 50 µL of 6 M guanidine HCl and revealed by adding 200 µL of Ellman's Reagent (0.2 mM DTNB (5,5-dithio-bis-(2-nitrobenzoic acid, Sigma), 50 mM Tris-HCl pH 8.0, 1 mM EDTA). After 10 min of additional incubation at room temperature, the absorbance was measured at 415 nm in Spectramax i3 microplate reader using SoftMax Pro 6.3 software (Molecular Devices). Since different substrates give various level of background, reactions without enzyme were performed for each substrate and their values were used as a blank. For Dpa enzyme kinetics, 50 µL of Ellman's Reagent without EDTA was layered over 50 µL of reactions containing 2 µM of enzyme, its mutated derivative, or enzyme pre-treated with EDTA. Reactions contained 0.5 mM acCoA, 10 mM 1,3-DAP and 10 mM EDTA when indicated. Once assembled, kinetics reactions were immediately transferred to microplate reader and absorbance was measured every 2 minutes for 1 hour. In order to convert the DTNB signal to the quantity of reaction products, standard curves were generated by measuring the signal of dilutions of CoA in the range of 0.01 to 1 mM. All assays were performed in triplicates and their average with standard deviations were plotted.

### Reporting summary

Further information on research design is available in the Nature Portfolio Reporting Summary linked to this article.

## Data availability

The final atomic model and coordinates of Dpa and of the Dpa in complex with the 1,3-DAP generated in this study have been deposited to the Protein Data Bank (PDB) under the accession codes 8A9O and 8A9N respectively. Previously reported structures used in this study are available in the PDB under accession codes: 4R9M, 3BJ7, 1S5K, 1QSO, 1TIQ, 4R87, 1QSM, 3BJ8, 5JPH, 4JLY, 2B5G. Uncropped gels, Western-blots and data generated in this study are provided in Source Data file. Source data are provided with this paper.

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

## Acknowledgements

This work was supported by a Research Grant 2022 from the European Society of Clinical Microbiology and Infectious Diseases (ESCMID) to DJ; the Fonds National de Recherche Scientifique (FNRS-MIS F4526.23) to DJ. The FRFS-WELBIO CR-2017S-03, FNRS CDR J.0068.19 and CDR J.0066.21, FNRS-EQP UN.025.19 and UN.031.N21F; and FNRS-PDR T.0066.18 and PDR T.0090.22 to AGP); ERC (CoG DiStRes, n° 864311 to AGP) and Joint Programming Initiative on Antimicrobial Resistance, (JPIAMR) JPI-EC-AMR-R.8004.18 to AGP; the Program Actions de Recherche Concerté 2016-2021, Fonds d'Encouragement à la Recherche of ULB (AGP); Fonds Jean Brachet and the Fondation Van Buuren (AGP); The authors acknowledge the use of beamtimes PROXIMA 1 and 2 A at the Soleil synchrotron (Gif-sur-Yvette, France). The authors thank Mykolas Bendorius, Beatričė Radavičiūtė and Paulė Želvytė for their involvement in the early steps of the project and Laurence Van Melderen for giving the possibility to develop part of the project in her lab.

## Author contributions

D.J., A.G-P., J.A., E.S. designed the research. D.J., J.A., A.Č., G.Š., J.M., J.S., C.M. performed research. J.A., E.S., C.M., A.G.P., D.J. provided tools. J.A., A.G.P., D.J. analyzed the data. D.J and A.G.P. wrote the manuscript with contributions from all the authors. All authors approved the final revision of the manuscript.

## Competing interests

The authors declare no competing interests.
