## [Peer Review File · Nature Communications]

A polyamine acetyltransferase regulates the motility and biofilm formation of *Acinetobacter baumannii*REVIEWER COMMENTS

Reviewer #1 (Remarks to the Author):

Armalyt  and colleagues present the identification and characterisation of a novel diaminopropane acetyltransferase (termed Dpa) responsible for acetylation of 1,3-diaminopropane (DAP) in *Acinetobacter baumannii*, an often multidrug-resistant nosocomial pathogen, for which novel drugs and drug targets are urgently sought-after. DAP has been previously shown to play a crucial role in virulence and motility in *A. baumannii* so that targeting the DAP regulon offers potential to combat the pathogen. In that, work to understand the DAP regulon is of utmost importance.

The authors convincingly demonstrate the acetylating activity of Dpa on DAP and by means of a knock-out mutant demonstrate the crucial role of Dpa in virulence, motility and biofilm formation, the latter being another key factor of pathogenesis and survival in the hospital environment.

Finally, they provide a comprehensive X-ray structural analysis of Dpa in complex with its co-substrate Acetyl-CoA and of a Dpa derivative impaired in catalytic activity allowing to capture a post-catalytic state with bound Acetyl-DAP.

Collectively, the topic is very important and the outcome presented is highly significant and novel. The story presented is well-written, straightforward and consistent and I cannot see any major flaw in the data.

Regarding the interpretation of the data and the provisioning of data and information to the public I have some points of criticism.

1. There is no information regarding the penetrance of the dpa gene within the species. At least, it should be stated if the gene is present in all 8 international clones and also in the type strain and other strains commonly used. Suitability as drug target requires high penetrance and can thus not be judged and discussed without this information.
2. I could not find a reference to the sequence of the dpa gene in the manuscript. This means that the interested reader needs to use the primer sequences provided to identify the locus. This is very inconvenient! Best, the authors should provide a deposit of the whole genome of the strain used. If this is unavailable, at least the sequence of the dpa gene with its vicinity should be provided to the public database together with the novel annotation of the gene.
3. The status of the structural data provision at PDB is "HOLD FOR RELEASE" so that I was unable to access and judge the data.

Miscellaneous:

4. Consider mentioning the catalytically inactive derivative in complex with the product in the abstract.
5. Line 37: "Acinetobacter, from the greek "a-kineto bakter" means non-motile rod". Please make sure that this is correct. It is true that 'Acinetobacter' means "non-motile rod" but I am unsure if it is correct to claim that "a-kineto bakter" is Greek; rather it appears to be derived from Greek words for movement and rod?!
6. Line 209: rephrase ...acCoA binds in a similarly in all...
7. Line 216: suppelementary → supplementary
8. Line 230: *A. baumanii* → *A. baumannii*
9. The prior annotation of the dpa genes as cheA has not be discussed. Is there a relation of DAP to chemotaxis?

Reviewer #2 (Remarks to the Author):

In this manuscript the authors report that they have discovered a unique acetylating enzyme that is only found in the bacterium, *A. baumannii*, which is a highly destructive hospital-acquired pathogen responsible for ventilator-associated pneumonia and sepsis. A unique polyamine, 1,3-diaminopropane (1,3-DAP), has been shown to be linked to surface-associated motility and virulence. The authors deduced that a unique acetyl-transferase must be expressed by *A. baumannii* in order to carry out the acetylation of 1,3-DAP. The authors identified the potential

gene sequence and demonstrated through a series of gene knock out, complementation studies and differential substrate specificity studies that the novel enzyme, Dpa, is the polyamine acetyltransferase. They also demonstrated that it is required for motility. The authors determined the crystal structure of Dpa and found similarities to other acetyl transferases but also significant differences. They were able to identify the active site and both the AcCoA and amine binding sites and characterize their amino acid composition. This information could be very useful in the development of potential drug lead compounds, since the gene is essential for biofilm formation, significantly different from other acetyl-transferases and given the current complications from pneumonia due to SARs-CoV2 infections of timely interest. I recommend acceptance once the following criticisms are addressed.

- 1) The authors do not offer an explanation for the observed substrate specificity. This should be included.
- 2) The authors should demonstrate if the products are mono- or bis- acetylated.
- 3) What sequence data do the authors have demonstrating that the target genes are indeed KO?

Reviewer #3 (Remarks to the Author):

The manuscript by J. Armalyte et al. presents the crystal structure of novel polyamine acetyltransferase Dpa from highly resistant hospital-acquired pathogen *Acinetobacter baumannii* that causes infections in the lungs, blood, urinary tract, and open wounds. This study reveals that Dpa acetylates short polyamine 1,3-diaminopropane (1,3-DAP), directly controlling cell motility and biofilm formation. Crystal structures of Dpa in complex with cofactor and ternary complex with products (coenzyme A and acetyl-DAP) were determined at high resolution. It is shown that Dpa can also acetylate other polyamines such as putrescine, spermidine and spermine (Figure 2b). However, the authors do not pay sufficient attention to this interesting fact. Spermine, spermidine and putrescine are present in mammalian cells and might be associated with *A. baumannii* virulence and antibiotic resistance. It is known that spermine and spermidine are natural substrates for AmvA multidrug efflux pump from *A. baumannii* that could export these polyamines from the external environment. Thus, the authors do not take into account that Dpa could regulate concentrations of spermidine and spermine within the bacterial cells.

The authors discuss that Dpa is the first polyamine acetyltransferase described in *A. baumannii*. In the same section (Discussion), the authors admit that "Dpa structurally closer to the well-known eukaryotic acetyltransferases SSAT and HPA2 that acetylate polyamines." Sequences and structures between Dpa and HPA2 were not compared. Moreover, HPA2 from *A. baumannii* has been described previously (J. S. Tomar and R. V. Hosur, 2020). Interestingly, HPA2 from *A. baumannii* belongs to GNAT acetyltransferase superfamily and could acetylate a wide range of substrates, including polyamines (spermidine and spermine), histones, and antibiotics. Consequently, additional sequence and structure comparison analysis between Dpa and other known prokaryotic and eukaryotic polyamine acetyltransferases from GNAT family would be important to carry out in order to investigate the function of Dpa. In addition to SpeG and SSAT, there are other known polyamine acetyltransferases such as BltD from *Bacillus subtilis*, PaiA from *Bacillus subtilis* (structure is known), SSAT from *Mus musculus* (structure is known), HPA2 from *Homo sapiens* (structure is known) and HPA2 from *A. baumannii* that should be included into the analysis. The observed homology between Dpa, SpeG and SSAT suggests that Dpa is an unusual acetyltransferase with a different function. However, additional analysis would be needed to support this conclusion.

The structure of Dpa "revealed a dimeric state with a conserved acetyl-CoA binding site." The authors should consider investigating the oligomeric state of Dpa in solution in the presence of cofactor and/or other polyamines. For example, SpeG and HPA2 could adopt different homo-oligomeric states depending on the present substrate and its concentration. Oligomeric state of acetyltransferases is often associated with their acetylation mechanism.

Remarkably, the structure of the catalytically inactive enzyme with the substitution of the important tyrosine shows the presence of acetylated 1,3-DAP in the potential polyamine binding site. How could authors explain the acetylated polyamine's presence in the determined structure? It is possible that acetyl-DAP is an artifact of soaking experiments (used at concentration 2M) and might be physiologically not relevant. Did authors soak crystals of Dpa in a solution containing 2M

1,3-DAP for several seconds (as described in methods) or 30 seconds to 1 minute (as described in results)? Although, it is not clear if the authors observe acetylated substrate in one monomer or both monomers of the Dpa dimer. Are there any differences in conformation between dimer of Dpa in complex with acetyl-CoA and dimer of Dpa in complex with acetylated 1,3-DAP and cofactor? The role of Mg²⁺ ion in the structure is not clear. The coordination geometry of Mg²⁺ ion is also questionable (Figure 3e).

My overall impression is that the function of Dpa is not fully assessed. Additional in vivo and in vitro analysis to investigate the effect of other polyamines (spermidine and spermine) on motility and biofilm formation compared to 1,3-DAP would be important in order to understand Dpa function in *A. baumannii*. I believe the present analysis is insufficient to qualify this paper for publication in Nature Communication.

RESPONSE TO REVIEWER COMMENTS (NCOMMS-22-33920)

Reviewer #1 (Remarks to the Author):

Armalytè and colleagues present the identification and characterisation of a novel diaminopropane acetyltransferase (termed Dpa) responsible for acetylation of 1,3-diaminopropane (DAP) in *Acinetobacter baumannii*, an often multidrug-resistant nosocomial pathogen, for which novel drugs and drug targets are urgently sought-after. DAP has been previously shown to play a crucial role in virulence and motility in *A. baumannii* so that targeting the DAP regulon offers potential to combat the pathogen. In that, work to understand the DAP regulon is of utmost importance.

The authors convincingly demonstrate the acetylating activity of Dpa on DAP and by means of a knock-out mutant demonstrate the crucial role of Dpa in virulence, motility and biofilm formation, the latter being another key factor of pathogenesis and survival in the hospital environment. Finally, they provide a comprehensive X-ray structural analysis of Dpa in complex with its co-substrate Acetyl-CoA and of a Dpa derivative impaired in catalytic activity allowing to capture a post-catalytic state with bound Acetyl-DAP.

Collectively, the topic is very important and the outcome presented is highly significant and novel. The story presented is well-written, straightforward and consistent and I cannot see any major flaw in the data.

Regarding the interpretation of the data and the provisioning of data and information to the public I have some points of criticism.

We thank the reviewer for his enthusiasm about our manuscript. We appreciate the suggested improvements that will make the information access easier.

1. There is no information regarding the penetrance of the *dpa* gene within the species. At least, it should be stated if the gene is present in all 8 international clones and also in the type strain and other strains commonly used. Suitability as drug target requires high penetrance and can thus not be judged and discussed without this information.

To answer this question, we have downloaded all accessible *A. baumannii* strains from NCBI database (516 genomes as of December 2022) and performed blast search of the *dpa* gene. We have found it to be present in 100 % of the strains and conserved with more than 95% identity. The only case where conservation was 82% seems to be a species typing error, since that strain using automated methods would be assigned as *A. pittii*. In conclusion, *dpa* is highly conserved and could be considered as a part of the core genome of *A.*

baumannii. In the revised manuscript we have added this information, line 87 now reads:

“BLASTN analysis of available *A. baumannii* strains (515 as of December 2022) showed that *dpa* gene was present in all sequenced strains and its nucleotide sequence was conserved with more than 95 % identity.”

2. I could not find a reference to the sequence of the *dpa* gene in the manuscript. This means that the interested reader needs to use the primer sequences provided to identify the locus. This is very inconvenient! Best, the authors should provide a deposit of the whole genome of the strain used. If this is unavailable, at least the sequence of the *dpa* gene with its vicinity should be provided to the public database together with the novel annotation of the gene.

We agree with the reviewer’s comment. We have submitted a gene locus of *dpa* to GenBank (accession number OQ718427) and provided the identifier in the methods section. Since in the revised version of the manuscript now includes comparison of Dpa activity to other acetyltransferases we have added a table (Supplementary Table S3) with amino acid sequences of the enzymes used in this study in order to avoid any confusion. Please note, that protein sequence will also become intrinsically linked to the structure and to the article, once PDB deposition is released. This will facilitate the identification of Dpa and its homologues.

3. The status of the structural data provision at PDB is “HOLD FOR RELEASE” so that I was unable to access and judge the data.

This is a standard procedure for structural data submitted for publication. The coordinates and structure factors become available upon publication. The release will be programmed before the publishing date.

Miscellaneous:

4. Consider mentioning the catalytically inactive derivative in complex with the product in the abstract.

Thank you for suggestion. The last part of the abstract now reads:

“Structure of catalytically impaired derivative of Dpa in complex with the reaction product shows that binding and orientation of the polyamine substrates is conserved between different polyamine-acetyltransferases.”

5. Line 37: “Acinetobacter, from the greek “a-kineto bakter” means non-motile rod”. Please make sure that this is correct. It is true that ‘Acinetobacter’ means

“non-motile rod” but I am unsure if it is correct to claim that “a-kineto bakter” is Greek; rather it appears to be derived from Greek words for movement and rod?!

Thank you for the comment, this has been clarified and now reads:
“The name *Acinetobacter* is derived from the greek words motion (“kineto”) and rod (“bakter”) and means non-motile bacterium.”

6. Line 209: rephrase ...acCoA binds in a similarly in all...

We apologize for the typing mistake. The phrase has been corrected and now reads:

“Overall, acCoA binding is similar in all GNATs”

7. Line 216: suppelementary → supplementary

Corrected.

8. Line 230: A. baumanii → A. baumannii

Typing of the species name has been crosschecked and corrected in the manuscript.

9. The prior annotation of the dpa genes as cheA has not be discussed. Is there a relation of DAP to chemotaxis?

No, there was no previous relation to chemotaxis. CheA was proposed to be part of the type II toxin antitoxin system with preceding gene CheT (PMID: 23667234) named after “switched-element toxin-antitoxin system”, however our following tests have shown lack of evidence for this function. We have found that the two proteins do not interact and are functionally uncoupled, and that CheA was an acetyltransferase functionally and structurally unrelated to recently described acetyltransferase toxins that modify tRNA. CheT seems to be a relic of toxin-antitoxin couple, since homologous genes outside of the *Acinetobacter* genus are located next to RelE toxins. To avoid confusion, we have not discussed this previous relation in the manuscript and we focused on the function the enzyme that we discovered.

Reviewer #2 (Remarks to the Author):

In this manuscript the authors report that they have discovered a unique acetylating enzyme that is only found in the bacterium, *A. baumannii*, which is

a highly destructive hospital-acquired pathogen responsible for ventilator-associated pneumonia and sepsis. A unique polyamine, 1,3-diaminopropane (1,3-DAP), has been shown to be linked to surface-associated motility and virulence. The authors deduced that a unique acetyl-transferase must be expressed by *A. baumannii* in order to carry out the acetylation of 1,3-DAP. The authors identified the potential gene sequence and demonstrated through a series of gene knock out, complementation studies and differential substrate specificity studies that the novel enzyme, Dpa, is the polyamine acetyl-transferase. They also demonstrated that it is required for motility. The authors determined the crystal structure of Dpa and found similarities to other acetyl transferases but also significant differences. They were able to identify the active site and both the AcCoA and amine binding sites and characterize their amino acid composition. This information could be very useful in the development of potential drug lead compounds, since the gene is essential for biofilm formation, significantly different from other acetyl-transferases and given the current complications from pneumonia due to SARs-CoV2 infections of timely interest. I recommend acceptance once the following criticisms are addressed.

We thank the reviewer for his positive opinion about our manuscript and valuable criticisms that we have tried to answer in the revised version.

1) The authors do not offer an explanation for the observed substrate specificity. This should be included.

Following the remarks of reviewer #3 (see below), we have now included broad comparison of the substrate specificity with other bacterial acetyltransferases – *Escherichia coli* SpeG, a previously reported *A. baumannii* GNAT family acetyltransferase named Hpa2 and aminoglycoside acetyltransferase Aac(6') from *Salmonella enterica*. All enzymes were compared under identical conditions for acetylation of polyamines, aminoglycosides and amino acids. This comparison led to many valuable conclusions, most importantly that Dpa is most specific to 1,3-DAP; that other GNAT family acetyltransferases (SpeG and Aac(6')) acetylate larger polyamines or amino acids (Hpa2 and SpeG) or aminoglycoside antibiotics (Aac(6')).

We have discussed the points regarding the substrate specificity in a new separate chapter. The results on different substrates are presented in new Figure 2c, Supplementary Figure S4 and S5. In depth structural comparisons have been performed as requested by the reviewer #3 and are summarized in new Figure 4, Figure 5 and Supplementary Figure S6. We believe that we have now clearly indicated structural elements (β 3- β 4 turn and acidic residues located in this region, as well as C terminal beta strands) likely to be responsible for substrate specificity as well as for oligomeric state.

2) The authors should demonstrate if the products are mono- or bis- acetylated.

We agree with the remark of the reviewer. In order to determine whether the 1,3-DAP can be mono or di-acetylated we have attempted acetylation of mono-acetylated DAP or chemically speaking, N-(3-aminopropyl)-acetamide. We have found that N-(3-aminopropyl)-acetamide was a poor substrate for Dpa and hence the major product of the reaction is mono-acetylated-DAP (now part of the Figure 2 d). Mono acetylated DAP also did not visibly inhibit the enzyme activity.

We have additionally performed mass spectrometry analysis of the enzymatic product to address this issue, however the spectra were quite difficult to interpret as polyamines are not easily visible without additional chemical modifications that typically use amine groups. Addition of enzyme also introduced additional peaks coming from buffer or sample preparation. Nevertheless, we could detect the mono-acetylated, but not di-acetylated reaction products (see below).

3) What sequence data do the authors have demonstrating that the target genes are indeed KO?

The region around the deletion zone (550 bp) was amplified by PCR and clean deletion has been confirmed by Sanger sequencing.

Reviewer #3 (Remarks to the Author):

The manuscript by J. Armalyte et al. presents the crystal structure of novel polyamine acetyltransferase Dpa from highly resistant hospital-acquired pathogen *Acinetobacter baumannii* that causes infections in the lungs, blood, urinary tract, and open wounds. This study reveals that Dpa acetylates short polyamine 1,3-diaminopropane (1,3-DAP), directly controlling cell motility and biofilm formation. Crystal structures of Dpa in complex with cofactor and ternary complex with products (coenzyme A and acetyl-DAP) were determined at high resolution. It is shown that Dpa can also acetylate other polyamines such as putrescine, spermidine and spermine (Figure 2b). However, the authors do not pay sufficient attention to this interesting fact. Spermine, spermidine and putrescine are present in mammalian cells and might be associated with *A. baumannii* virulence and antibiotic resistance. It is known that spermine and spermidine are natural substrates for AmvA multidrug efflux pump from *A. baumannii* that could export these polyamines from the external environment. Thus, the authors do not take into account that Dpa could regulate concentrations of spermidine and spermine within the bacterial cells.

We thank the reviewer for his insights on the potential role of Dpa in virulence by acetylating other polyamines present in mammalian cells. Indeed, we have demonstrated *in vitro* that other polyamines can be acetylated by Dpa (Figure 2). We have now discussed this possibility in the text (lines 136-137) and performed additional experiments. We have found that addition of other polyamines globally did not have significant impact on motility and biofilm formation profiles or was not dependent on presence of *dpa* gene in case of spermine that inhibited motility (lines 135-136 in the text and Supplementary Figure S2).

We have also attempted structural analysis of Dpa bound to other polyamines. Unfortunately, although we did not succeed in producing meaningful diffraction data that allowed structure determination of such complexes.

The authors discuss that Dpa is the first polyamine acetyltransferase described in *A. baumannii*. In the same section (Discussion), the authors admit that “Dpa structurally closer to the well-known eukaryotic acetyltransferases SSAT and HPA2 that acetylate polyamines.” Sequences and structures between Dpa and

HPA2 were not compared. Moreover, HPA2 from *A. baumannii* has been described previously (J. S. Tomar and R. V. Hosur, 2020). Interestingly, HPA2 from *A. baumannii* belongs to GNAT acetyltransferase superfamily and could acetylate a wide range of substrates, including polyamines (spermidine and spermine), histones, and antibiotics.

We thank the reviewer for this remark. Indeed, Hpa2 in *A. baumannii* was first described bioinformatically (PMID: 27125865) and named based on proposed similarity to the yeast Hpa2 and *in silico* docking of polyamines. This publication also suggested that the enzyme is similar to Aac(6') from *Salmonella* that acetylates antibiotics and histones (PMID: 15123251). Later, it was shown that Hpa2_{Ab} conferred resistance to aminoglycosides (PMID: 30573651). It was also shown *in vitro* to modify spermine, spermidine and to some extent putrescine (PMID: 31786230). In order to compare the Hpa2 activity to Dpa, we have cloned and purified this enzyme and our tests have shown virtually no acetylation of polyamines, nor the antibiotics (new Figure 2C and new Supplementary Figure S4). Additionally, since Hpa2_{Ab} and Dpa did not confer resistance to aminoglycosides as previously suggested, we have cloned *Salmonella enterica* Aac(6') enzyme as a control. As reported in the literature, Aac(6') could provide resistance to kanamycin, gentamicin, amikacin and tobramycin (Supplementary Figure S4). Nevertheless, we showed that the purified *A. baumannii* Hpa2 was active on other substrates. Modelling with AlphaFold2 provided a highly confident model with TM-score (pTM = 0.9) and structural homologue search with DALI sever retrieved a very good match with SACOL1063 from *Staphylococcus aureus* (Z score 24.0, rmsd 1.2). SACOL1063 acetyltransferase was demonstrated to modify amino acids (PMID: 27783928) and we have therefore tested the Hpa2 as well as Dpa, SpeG and Aac(6') for acetylation of all amino acids. We have found that Hpa2_{Ab} acetylates Threonine, and to some extent Tryptophan, Tyrosine, Arginine, Serine, Histidine. To our surprise, SpeG could also acetylate the same amino acids but its activity was much lower (new Supplementary Figure S5). Comparisons of substrate specificities are not summarized in a new paragraph. As requested further by the reviewer we have included thorough structural comparison of Dpa with (as well as *A. baumannii* Hpa2 model) with other acetyltransferases (new Figures 4, 5 and Supplementary Figure S6). In fact, our comparisons show that Dpa is a closer homologue to yeast Hpa2 than is the Hpa2_{Ab}. These new data on *A. baumannii* Hpa2 is summarized in Supplementary Figure S5.

Consequently, additional sequence and structure comparison analysis between Dpa and other known prokaryotic and eukaryotic polyamine acetyltransferases from GNAT family would be important to carry out in order to investigate the function of Dpa. In addition to SpeG and SSAT, there are other known

polyamine acetyltransferases such as BltD from *Bacillus subtilis*, PaiA from *Bacillus subtilis* (structure is known), SSAT from *Mus musculus* (structure is known), HPA2 from *Homo sapiens* (structure is known) and HPA2 from *A. baumannii* that should be included into the analysis. The observed homology between Dpa, SpeG and SSAT suggests that Dpa is an unusual acetyltransferase with a different function. However, additional analysis would be needed to support this conclusion.

We agree with the reviewer, as mentioned before, we provide now in the revised version further sequence and structure (or model) alignments with all the mentioned acetyltransferases in context of oligomers (new Figure 4) and monomers (new Supplementary Figure 6) as well as their sequence alignments (Supplementary Figure S6). Most importantly, we also provide substrate specificity comparisons between the four bacterial acetyltransferases Dpa and SpeG, as well as Hpa2 and Aac(6'). We found that Dpa was mostly specific for 1,3-DAP and that, on the contrary, SpeG acetylates long polyamines, and its activity decreases with shorter polyamines (this is now part of Figure 2 c).

The structure of Dpa “revealed a dimeric state with a conserved acetyl-CoA binding site.” The authors should consider investigating the oligomeric state of Dpa in solution in the presence of cofactor and/or other polyamines. For example, SpeG and HPA2 could adopt different homo-oligomeric states depending on the present substrate and its concentration. Oligomeric state of acetyltransferases is often associated with their acetylation mechanism.

We thank the reviewer for this comment. We have now assessed the oligomeric state of Dpa in solution alone or in presence of acetyl-CoA and/or polyamines (new Supplementary Figure S7). In all conditions tested we have found that Dpa remained dimeric, which can be explained by extensive interactions between monomers supported by β -strand exchange. For control, we have tested *A. baumannii* Hpa2, however its retention time in size exclusion experiments suggested monomeric state, which is also supported by AlphaFold model (ptm of dimer 0.69 vs 0.9 for monomer, and iptm = 0.46). Unfortunately, previous studies on Hpa2Ab did not include the AcCoA-only control which we demonstrate to be also visible under 280 nm (see AcCoA control in Supplementary Figure S7 c) and thus appearance of second peak after addition of AcCoA corresponds to the retention time and amount and AcCoA (Supplementary Figure S7d, f, h).

Remarkably, the structure of the catalytically inactive enzyme with the substitution of the important tyrosine shows the presence of acetylated 1,3-DAP in the potential polyamine binding site. How could authors explain the acetylated polyamine's presence in the determined structure? It is possible that

acetyl-DAP is an artifact of soaking experiments (used at concentration 2M) and might be physiologically not relevant. Did authors soak crystals of Dpa in a solution containing 2M 1,3-DAP for several seconds (as described in methods) or 30 seconds to 1 minute (as described in results)? Although, it is not clear if the authors observe acetylated substrate in one monomer or both monomers of the Dpa dimer. Are there any differences in conformation between dimer of Dpa in complex with acetyl-CoA and dimer of Dpa in complex with acetylated 1,3-DAP and cofactor?

We understand the concerns of the reviewer. In order to observe the acetylation “in action” we have used catalytically impaired, but still active Dpa enzyme. We have now provided the acetylation kinetics curve for Y128F mutant (Supplementary Figure S8). Due to this residual activity and high concentration of 1,3-DAP used, the acetylation process is slower and allowed us to observe the substrate in pre- and post-catalysis but still in the active site. We did not observe additional densities apart from the active site, which together with substrate specificity tests indicate that the observed binding in the structure is most likely correct. Moreover, the polyamine was located in the channel leading to AcCoA and at the position similar to that previously described for other GNAT enzymes. Soaking of crystals with substrates is a well-established strategy in structural biology resulting in the determination of many structures of enzymes bound to their product post-catalysis (these examples will be provided as reference in the revised version, e.g. Tamman et al., *Nature Chem Biol.* 2020; Xiao et al., *Nature Struct and Mol Biol.* 2010; Hogg et al., *Cell* 2004). We apologize for confusion about the time of soaking, this has been clarified in methods.

We agree that the difference between different monomers was not clear. Indeed, 1,3-DAP was present in both monomers – in one case it was not yet acetylated, in another case it was already acetylated. To improve clarity, this has been addressed in text and we have additionally provided alignments of individual monomers to clarify that no major structural differences are observed in the dimer (Supplementary Figure S9).

The role of Mg²⁺ ion in the structure is not clear. The coordination geometry of Mg²⁺ ion is also questionable (Figure 3e).

We thank the reviewer for raising this issue. While we cannot unambiguously claim the identity of the density of this metal ion, we have modified the text accordingly. In order to clarify the role of presumably Mg²⁺ (or other metal) ion, we have performed the enzyme kinetics with the enzyme pre-treated with EDTA and we have not found any substantial differences (Supplementary Figure S8) and we have commented on this fact in the text (lines 215-216).

My overall impression is that the function of Dpa is not fully assessed. Additional *in vivo* and *in vitro* analysis to investigate the effect of other polyamines (spermidine and spermine) on motility and biofilm formation compared to 1,3-DAP would be important in order to understand Dpa function in *A. baumannii*. I believe the present analysis is insufficient to qualify this paper for publication in Nature Communication.

We thank the reviewer for thorough criticisms, as requested by the reviewer we have performed experiments and analysis on all the questions raised. Additional *in vivo* experiments showed that the motility and biofilm formation was dependent on *dap* in relation to 1,3-DAP but not the other polyamines. We have additionally clarified *in vitro* that Dap is a stable dimer in solution independently of the presence of substrates, and that the product of the reaction is mono-acetyl-diaminopropane (N-(3-aminopropyl)acetamide). We have now included other acetyltransferases in our study - a well described spermine-spermidine acetyltransferase SpeG from *E. coli*, a previously reported Hpa2 acetyltransferase from *A. baumannii* and an aminoglycoside acetyltransferase Aac(6') from *S. enterica*. We have compared the activity of all these enzymes on polyamines, aminoglycosides and amino acids which helped to clarify the role and substrate specificity and compare structures of all of them. We have provided systematic structural alignments of all enzymes tested and those indicated by the reviewer. Finally, we have described the key features that are most likely responsible for oligomeric state and substrate binding.

REVIEWERS' COMMENTS

Reviewer #1 (Remarks to the Author):

As already stated in my original review, I consider this an important piece of work. I am happy with the responses and amendments in answer to my points of criticism. Moreover, I found the responses to the other reviewers' comments appropriate and I was happy to see the many additional valuable information supplemented in response to all comments. I fully support acceptance of this work.

Reviewer #2 (Remarks to the Author):

The authors have addressed my criticisms. The manuscript is now acceptable for publication.

Reviewer #3 (Remarks to the Author):

The manuscript entitled "A polyamine acetyltransferase regulates the motility and biofilm formation of *Acinetobacter baumannii*" has been carefully revised by Dr Dukas and co-authors. Authors addressed my comments concerning the Dpa function, sequence and structure analysis of the Dpa with known polyamine acetyltransferases, clarified presents of acetylated 1,3-DAP at the binding site. Importantly authors provided additional kinetic analysis of different polyamine acetyltransferases including SpeG from *E. coli*, Hpa2 acetyltransferase from *A. baumannii* and Aac(6') acetyltransferase from *Salmonella* to clarify Dpa substrate specificity. In addition, authors addressed my comment about oligomeric state of the Dpa in solution in the presence of different substrates. The information about crystal soaking experiments was revised. I believe that revised work resulted in an improved manuscript. I am satisfied with authors responses to my comments and questions.